# FlexiClip: Locality-Preserving Free-Form Character Animation

**Anant Khandelwal** [1]

## Abstract

Animating clipart images with seamless motion while maintaining visual fidelity and temporal coherence presents significant challenges. Existing methods, such as AniClipart, effectively model spatial deformations but often fail to ensure smooth temporal transitions, resulting in artifacts like abrupt motions and geometric distortions. Similarly, text-to-video (T2V) and image-to-video (I2V) models struggle to handle clipart due to the mismatch in statistical properties between natural video and clipart styles. This paper introduces FlexiClip, a novel approach designed to overcome these limitations by addressing the intertwined challenges of temporal consistency and geometric integrity. FlexiClip extends traditional Bézier curve-based trajectory modeling with key innovations: temporal Jacobians to correct motion dynamics incrementally, continuous-time modeling via probability flow ODEs (pfODEs) to mitigate temporal noise, and a flow matching loss inspired by GFlowNet principles to optimize smooth motion transitions. These enhancements ensure coherent animations across complex scenarios involving rapid movements and non-rigid deformations. Extensive experiments validate the effectiveness of FlexiClip in generating animations that are not only smooth and natural but also structurally consistent across diverse clipart types, including humans and animals. By integrating spatial and temporal modeling with pre-trained video diffusion models, FlexiClip sets a new standard for high-quality clipart animation, offering robust performance across a wide range of visual content. Project Page: https://creative-gen.github.io/flexiclip.github.io/

## 1. Introduction

Animating static clipart images while preserving their visual integrity and ensuring temporal coherence in motion is a challenging problem in computer graphics. Existing methods, like AniClipart (Wu et al., 2024) address this by modeling key point trajectories with cubic Bézier curves and applying ARAP deformation to maintain geometric consistency. Gal23 (Gal et al., 2024) is also a similar work learning neural displacement field with pre-trained T2V (Text-to-Video) diffusion model on cubic Bézier curves. While AniClipart effectively captures spatial deformations, it struggles with maintaining temporal consistency across frames. Specifically, the method suffers from abrupt transitions, geometric distortions, and inconsistencies when generating complex motions or handling rapid pose transitions (Fig. 3). These artifacts arise from a rigid parametrization of the motion process that does not fully account for temporal noise or its correction over time. Additionally, recent T2V/I2V (Image-to-Video) models (Chen et al., 2023; 2024; Xing et al., 2025; Zhang et al., 2023; HaCohen et al., 2024; Wang et al., 2024; Lei et al., 2023) enable animation from text and images, but struggle to produce high-quality clipart animations due to the significant difference in statistical properties between natural videos and clipart.

To overcome these limitations, we introduce FlexiClip, a novel approach that extends the state of the art by addressing the key challenges of temporal coherence and geometric consistency in animated clipart. FlexiClip also builds on the basic framework of modeling keypoint trajectories with cubic Bézier curves but introduces significant innovations to improve temporal dynamics and maintain a consistent animation pipeline. Central to FlexiClip is the use of temporal Jacobian for incremental temporal corrections, pfODE (Lim et al., 2023; de Albuquerque & Pearson, 2024) for continuous-time integration of these corrections, and a flow matching loss inspired by GFlowNet (Bengio et al., 2023) to ensure smooth temporal evolution and reduction of temporal noise.

Although AniClipart also represents keypoint motion through Bézier curves but employs ARAP (As Rigid As Possible) deformation to model spatial consistency, its temporal modeling lacks a mechanism for addressing noise accumulation across frames. In contrast, FlexiClip intro-

[1]Search Technology Center India, Microsoft, IDC, Bengaluru, India(Bharat). Correspondence to: Anant Khandelwal <anantk@microsoft.com, anant.iitd.2085@gmail.com>.

*Proceedings of the 42nd International Conference on Machine Learning*, Vancouver, Canada. PMLR 267, 2025. Copyright 2025 by the author(s).

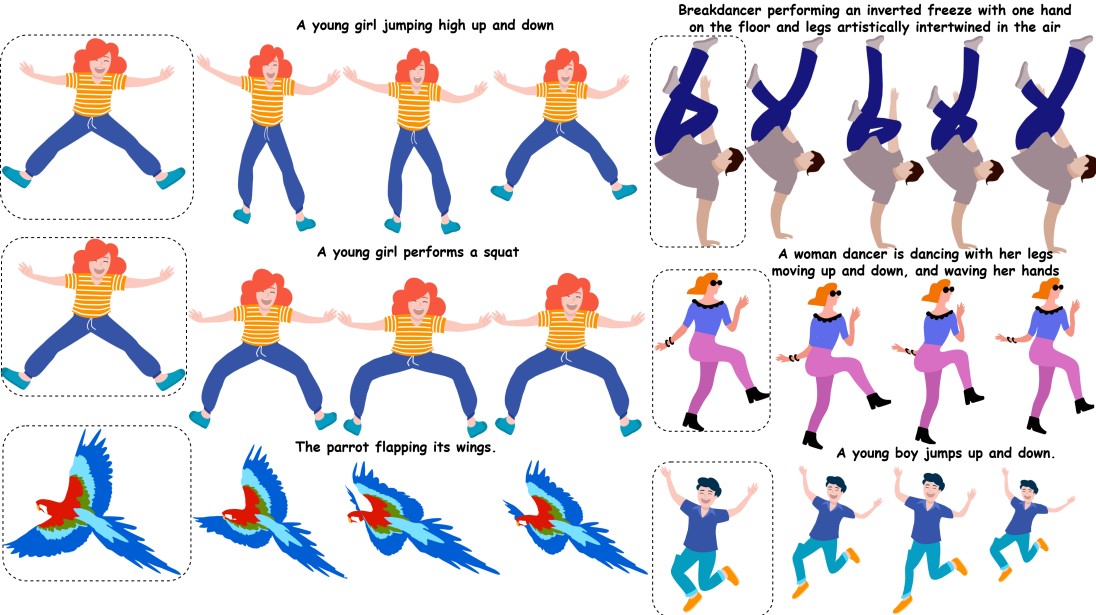

Figure 1. FlexiClip generates high-quality clipart animations based on text prompts, ensuring visual consistency and smooth temporal motion. Original clipart images are outlined with dashed boxes.

duces a novel paradigm by learning temporal Jacobians that incrementally correct the spatial Jacobian over time. This framework enables precise control over temporal evolution and prevents drift in the motion dynamics, thereby maintaining the animation's naturalness and consistency across longer sequences. Additionally, we leverage probability flow ODE (pfODE) (Sec.2.2) to model the temporal correction process as a continuous-time function, addressing temporal noise more effectively than discrete-time optimization methods.

A critical challenge in temporal animation is the preservation of motion smoothness without introducing geometric distortions. In previous works, such as AniClipart, motion dynamics are modeled independently for each frame, which can lead to inconsistent transitions and visual artifacts (Fig 3). In contrast, FlexiClip utilizes pfODE (Sec.2.2) to model the evolution of temporal Jacobian, which are corrected progressively over time. This continuous correction mechanism ensures that the temporal noise is mitigated, resulting in smoother and more consistent motion transitions. Moreover, this framework provides a novel solution to the problem of maintaining structural consistency (locality preserving) during fast spatial transitions, which is often a difficult task in motion modeling.

Another novel contribution of FlexiClip is the flow matching loss, which leverages the principles of GFlowNet (Bengio et al., 2023) to optimize the temporal noise reduction process. The flow matching loss operates by comparing the dynamics of the spatial Jacobian and their temporal corrections over time. This approach ensures that temporal

noise introduced by the spatial model is progressively reduced, facilitating smoother transitions between frames and preventing the accumulation of errors. Unlike earlier approaches, such as AniClipart, that do not explicitly model the evolution of temporal noise, FlexiClip provides a robust mechanism for controlling noise accumulation and preserving the underlying motion's coherence.

At last, FlexiClip integrates learning of spatial and temporal modules with the concept of video Score Distillation Sampling (SDS), which allows us to distill the knowledge of pre-trained video diffusion models to guide the animation generation process. Extensive experiments and ablation studies demonstrate FlexiClip's ability to generate smooth, natural, and temporally consistent clipart animations across a wide range of visual content, including humans and animals (see Fig. 1). FlexiClip also supports handling complex animations (Fig. 5) where keypoint transitions involve nonrigid deformations like rotation, complex motion, etc. In summary, our contributions can be summarized as follows:

- **Coherent Temporal Corrections**: We introduce the novel concept of temporal Jacobians, which incrementally adjust the spatial geometry over time to account for temporal variations. This mechanism ensures smoother and more temporally coherent animations by addressing the issue of noise accumulation across frames.

- **Locality Preserving Deformation**: We propose the use of pfODE to model the continuous-time evolution of the temporal Jacobians. This approach allows

for more precise temporal corrections and ensures smoother transitions between frames, improving upon discrete-time methods such as those used in AniClipart.

- **Flow Matching Loss for Temporal Noise Reduction**: We introduce the flow matching loss, inspired by the GFlowNet framework, to optimize the reduction of temporal noise. This loss function ensures that the accumulated temporal noise is progressively reduced over time, leading to smoother, more consistent animations.

## 2. Preliminaries

### 2.1. Representing Shapes as Jacobian Fields

Let $\mathcal{M}_0 = (\mathbf{V}_0, \mathbf{F}_0)$ denote the initial mesh, where $\mathbf{V}_0 \in \mathbb{R}^{V \times 2}$ specifies the 2D vertex positions and $\mathbf{F}_0 \in \mathbb{R}^{F \times 2}$ describes the triangular faces. Keypoints are defined using an indicator matrix $\mathbf{K}_c \in \{0, 1\}^{V_c \times V}$, with the corresponding vertices represented as $\mathbf{V}_c = \mathbf{K}_c \mathbf{V}_0$. These keypoints are assigned target positions $\mathbf{T}_c = \mathbf{V}_c + \mathbf{D}_c$, where $\mathbf{D}_c \in \mathbb{R}^{V_c \times 2}$ defines the displacements.

The deformation of the mesh is characterized by a Jacobian field $\mathbf{J}_0 = \{\mathbf{J}_{0,f} \mid f \in \mathbf{F}_0\}$, where each face $f$ has an associated Jacobian matrix $\mathbf{J}_{0,f} \in \mathbb{R}^{2 \times 2}$. This matrix is computed as $\mathbf{J}_{0,f} = \nabla_f \mathbf{V}_0$, which represents the gradient of the vertex positions over the triangle $f$. To compute the deformed mesh $\mathbf{V}^*$, we solve the following optimization problem:

$$\mathbf{V}^* = \arg\min_{\mathbf{V}} \|\mathbf{LV} - \nabla^T \mathcal{A} \mathbf{J}\|^2, \quad (1)$$

where $\mathbf{L}$ is the cotangent Laplacian operator, $\mathcal{A}$ is the mass matrix, and $\mathbf{J}$ is the specified Jacobian field. To avoid trivial deformations such as global translations, constraints are applied to the keypoints, resulting in the following extended formulation:

$$\mathbf{V}^* = \arg\min_{\mathbf{V}} \|\mathbf{LV} - \nabla^T \mathcal{A} \mathbf{J}\|^2 + \lambda \|\mathbf{K}_c \mathbf{V} - \mathbf{T}_c\|^2, \quad (2)$$

where $\lambda > 0$ balances the influence of the constraint term. The solution is obtained by solving the linear system:

$$(\mathbf{L}^T \mathbf{L} + \lambda \mathbf{K}_c^T \mathbf{K}_c) \mathbf{V} = \mathbf{L}^T \nabla^T \mathcal{A} \mathbf{J} + \lambda \mathbf{K}_c^T \mathbf{T}_c, \quad (3)$$

which can be efficiently solved using techniques such as Cholesky decomposition. Let this be denoted with $g$ as a differentiable solver: $\mathbf{V}^* = g(\mathbf{J}, \mathbf{K}_c, \mathbf{T}_c)$. To learn the input shape $\mathbf{J}_0$ we set $\mathbf{D}_c = 0$ we learn it by minimizing the difference between the $\mathbf{J}$ and the identity, i.e., no deformation

$$L_0 = \sum_{i=1}^{|F|} \|\mathbf{J}_i - \mathbf{I}\| \quad (4)$$

Once the updated vertex positions $\mathbf{V}^*$ are computed, the corresponding updated Jacobian for each face $f$ is derived by taking the gradient of $\mathbf{V}^*$ over the face:

$$\mathbf{J}_f^* = \nabla_f \mathbf{V}^*. \quad (5)$$

For a triangular face $f$ with vertices $\{\mathbf{v}_i, \mathbf{v}_j, \mathbf{v}_k\}$, the Jacobian matrix captures how the vertex positions within the triangle are influenced by barycentric coordinates:

$$\mathbf{J}_f^* = \begin{bmatrix} \frac{\partial \mathbf{v}_i^*}{\partial \mathbf{v}_i} & \frac{\partial \mathbf{v}_j^*}{\partial \mathbf{v}_i} & \frac{\partial \mathbf{v}_k^*}{\partial \mathbf{v}_i} \\ \frac{\partial \mathbf{v}_i^*}{\partial \mathbf{v}_j} & \frac{\partial \mathbf{v}_j^*}{\partial \mathbf{v}_j} & \frac{\partial \mathbf{v}_k^*}{\partial \mathbf{v}_j} \\ \frac{\partial \mathbf{v}_i^*}{\partial \mathbf{v}_k} & \frac{\partial \mathbf{v}_j^*}{\partial \mathbf{v}_k} & \frac{\partial \mathbf{v}_k^*}{\partial \mathbf{v}_k} \end{bmatrix}. \quad (6)$$

Updated Jacobians give local deformation of each triangle.

### 2.2. Probability flow ODE (pfODE)

The diffusion process governs the evolution of data points over time via the stochastic differential equation (SDE):

$$dx = f(x, t)\, dt + G(x, t) \cdot dW, \quad (7)$$

where $f(x, t)$ is the drift term and $G(x, t)$ is the noise coefficient. Over time, the distribution transforms from $p_0(x)$ to an isotropic Gaussian distribution $p_T(x)$. To reverse this process, we model the reverse SDE as:

$$dx = \big(f(x, t) - \nabla \cdot [G(x, t)G(x, t)^T]$$
$$- G(x, t)G(x, t)^T \nabla_x \log p_t(x)\big)\, dt + G(x, t) \cdot d\bar{W}. \quad (8)$$

where $f(x, t)$ is the drift term, $G(x, t)$ represents the noise diffusion matrix, $\bar{W}$ is time-reversed Brownian motion and $\nabla_x \log p_t(x)$ is the score function, approximated by the diffusion model. This reverse process reconstructs the data distribution by denoising, guided by the score function $\nabla_x \log p_t(x)$. However, in addition to the SDE, Song et al. (Lim et al., 2023) proposed the probability flow ODE (pfODE), which satisfies the same Fokker-Planck equation, but is deterministic (no Brownian term). it is given by:

$$\frac{dx}{dt} = \bigg(f(x, t) - \frac{1}{2}\nabla \cdot [G(x, t)G(x, t)^T]$$
$$- \frac{1}{2}G(x, t)G(x, t)^T \nabla_x \log p_t(x)\bigg) \quad (9)$$

This pfODE evolves the data points smoothly without the stochastic Brownian motion term. This process is deterministic, and data points evolve smoothly, resulting in a flow that preserves local neighborhoods.

Under the Gaussian noise assumption, the score function $\nabla_x \log p_t(x)$ can be trained from a pre-trained diffusion model via score matching, and hence can guide mesh deformation starting with noisy vertices. However, the key

challenge in learning the mesh deformation directly with pre-trained diffusion models is that these models reverse the isotropic Gaussian noise, as described in the SDE-based formulation, making it harder to directly match the target distribution (highly structured, like posed mesh). It is straightforward to show (App. C) that the class of time-varying densities satisfies (9) when $f = 0$ and $GG^T = \dot{C}$, similar to (de Albuquerque & Pearson, 2024) we consider the pfODE with rescaling to avoid variance overflow:

$$\frac{d\tilde{x}}{dt} = A(t) \cdot \left( -\frac{1}{2}\dot{C}(t) \cdot \nabla_x \log p_t(x) \right) + \left( \dot{A}(t) \cdot A^{-1}(t) \right) \cdot \tilde{x}. \tag{10}$$

with $C(t)$ playing the role of injected noise and $A(t)$ the role of the scale schedule. Similar to (de Albuquerque & Pearson, 2024) we will leverage dimension-preserving flows for $C(t)$ and $A(t)$ to learn the exact temporal noise to be reversed. As the noise reduces, the rescaling $A(t)$ induces force in the opposite direction as the force induced by score function. This balance ensures that the distribution stabilizes asymptotically, maintaining local structure while evolving smoothly into the target distribution.

### 2.3. GFlowNets

Generative Flow Networks (GFlowNets) (Bengio et al., 2023) enable training generative models with unnormalized target densities. GFlowNet is represented as a directed acyclic graph $G = (S, A)$, where $S$ is the set of states and $A \subseteq S \times S$ is the set of actions. Transitions between states are deterministic, with an initial state $s_0$ and terminal states $s_N$. The forward policy $P_F(s'|s)$ defines the transition from $s$ to $s'$, while the backward policy $P_B(s|s')$ defines the reverse. The goal is to learn a forward policy such that the terminal state distribution $P_T(x) \propto R(x)$, where $R(x)$ is the unnormalized reward function.

The flow function $F(\tau)$ includes the normalizing factor, and $F(s)$ models the unnormalized probability flow for each state. Training uses the detailed balance (DB) condition, ensuring $F(s)P_F(s'|s) = F(s')P_B(s|s')$ for all transitions $(s \to s') \in A$. For terminal states $x$, the condition $F(x) = R(x)$ must hold. This ensures the terminal distribution $P_T(x)$ matches the desired target, proportional to $R(x)$.

Generative Flow Networks (GFlowNets) (Bengio et al., 2023) provide a framework for training generative models with an unnormalized target density function. A GFlowNet is represented as a directed acyclic graph $G = (S, A)$, where $S$ is the set of states and $A \subseteq S \times S$ is the set of actions. The transition between states is deterministic, and the network has an initial state $s_0$ and terminal states $s_N$. The forward policy $P_F(s'|s)$ defines the transition probability from state $s$ to $s'$, while the backward policy $P_B(s|s')$ defines the reverse transition. The goal is to learn a forward policy such that the terminal state distribution $P_T(x) \propto R(x)$, where

$R(x)$ is an unnormalized reward function.

The flow function $F(\tau)$ incorporates the normalizing factor, and the state flow function $F(s)$ models the unnormalized probability flow for each state. GFlowNet's training uses the detailed balance (DB) condition, which ensures that for any transition $(s \to s')$, the following holds: $F(s)P_F(s'|s) = F(s')P_B(s|s')$, $\forall(s \to s') \in A$. For terminal states $x$, the condition $F(x) = R(x)$ is required. Satisfying this DB criterion ensures that the terminal distribution $P_T(x)$ matches the desired target distribution, proportional to $R(x)$.

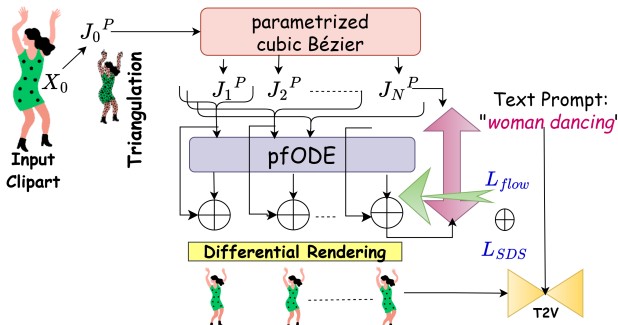

*Figure 2.* **System Design of FlexiClip**: A novel framework for generating temporally coherent and geometrically consistent animated clipart. FlexiClip leverages pfODE for continuous-time modeling on top of spatial posing, and a flow matching loss for reducing temporal noise, enabling smooth, natural animations across complex motion sequences.

## 3. FlexiClip

In this section, we introduce our mesh deformation framework, FlexiClip (Fig.2). We begin with a method overview (Sec. 3.1), followed by a spatial posing with Jacobian Fields on parameterized Bézier trajectories (Sec. 3.2). Next, we discuss how temporal signals are modeled using pfODE to ensure temporally coherent motion (Sec. 3.3). Finally, we outline the loss functions used (Sec. 3.4).

### 3.1. Method Overview

In FlexiClip, we detect keypoints and construct skeletons using UniPose (Yang et al., 2023) and skeleton generation (Cacciola, 2004), similar to AniClipart (Wu et al., 2024). Cubic Bézier trajectories define spatial motion, while pfODE and temporal Jacobian handle temporal noise. Attention networks estimate $C(t)$ and $A(t)$, and flow matching from GFlowNets reduces temporal noise. Video SDS loss enables learning from the single image input.

### 3.2. Spatial Posing

We begin by describing the mathematical framework to animate a clipart image, $x_0$, characterized by $M$ keypoints

$\{p_0(i)\}_{i=0}^{M-1}$ and associated cubic Bézier trajectories parameterized by control points $\{c(i)\}_{i=0}^{M-1}$. These trajectories dictate the temporal evolution of the keypoints $\{p_t(i)\}_{i=0}^{M-1}$ over $N$ timesteps, where $t \in \{0, 1, \ldots, N-1\}$. At timestep $t$, keypoint positions $p_t(i)$ are sampled along Bézier trajectories $c(i)$ as: $p_t(i) = \sum_{j=0}^{3} B_j(u_t)c_j(i)$, where $u_t \in [0, 1]$ is the normalized time and $B_j(u_t) = \binom{3}{j}(1 - u_t)^{3-j}u_t^j$. These updated keypoint positions $\{p_t(i)\}$ anchor the clipart's deformation for $t > 0$ and hence they are used as anchor vertices $\mathbf{T}_c$ in (Eq.2) to compute the spatial deformation of the object geometry across discrete time steps. Before learning the deformation for $t > 0$, we fix the input shape $\mathbf{J}_0$ using (Eq.4). While this method efficiently models the spatial deformation, it often fails to maintain temporal coherence across frames. Prior research (Wu et al., 2024; Gal et al., 2024) highlights that predicting future Bézier control points at discrete timesteps frequently results in distorted object identities, loss of geometric consistency, and abrupt transitions between frames. Even with parameterized learning methods (Wu et al., 2024), such as those optimized using Score Distillation Sampling (SDS) loss, unseen pose configurations often result in jerky or unnatural motion dynamics. (Fig.3).

### 3.3. Temporal Smoothing

Lets denote the Jacobians for $N$ time steps obtained from spatial posing is denoted as $\{\mathbf{J}_0^P, \mathbf{J}_1^P, \mathbf{J}_2^P, \ldots, \mathbf{J}_{N-1}^P\}$ called as spatial Jacobians. We reformulate the temporal smoothing problem to predict temporal Jacobian $\mathbf{J}_t^R$ as corrective terms to spatial Jacobian $\mathbf{J}_t^P$. The total Jacobian at time $t$ is given by:

$$\mathbf{J}_t = \mathbf{J}_t^P + \mathbf{J}_t^R \tag{11}$$

This decomposition focuses on learning localized corrections, enabling precise control and mitigating drift through incremental adjustments.

**ODE Formulation:** Since the first frame is stationary and given as input, we set the first frame's temporal Jacobian as, $\mathbf{J}_0^R = \mathbf{0} \in \mathbb{R}^{F \times 2 \times 2}$. Furthermore, we are required to remove only temporal noise from spatial Jacobians and wanted to add only temporal correction term (temporal Jacobian), we propose to reformulate the ODE given in (Eq.10) given as:

$$\frac{d\mathbf{J}_t^R}{dt} = f_R(\mathbf{J}_0^P, C_W^P, C_{W-1}^R, t; \theta_R), \tag{12}$$

where $\mathbf{J}_0^P$ is the base Jacobian of the first frame, $C_W^P$ and $C_{W-1}^R$ are attention-encoded features of current-window pose predictions and past-window temporals, respectively. The function $f_R$, parameterized by $\theta_R$, outputs temporal corrections. The temporal Jacobian at time $t$, $\mathbf{J}_t^R$, is obtained

via integration:

$$\mathbf{J}_t^R = \mathbf{J}_0^R + \int_0^t f_R(\mathbf{J}_0^P, C_W^P, C_{W-1}^R, \tau; \theta_R)\, d\tau. \tag{13}$$

Here, $\mathbf{J}_0^R$ is initialized to zero for the first frame, ensuring a neutral starting state. Numerical methods such as Euler's method are used for integration. In comparison to (Eq.10) we model the time-varying noise $C(t)$ with $C_W^P$ and rescaling $A(t)$ with $C_{W-1}^R$. $C_W^P$ is the attention over spatial Jacobians over a time window $W$ i.e. till the current time step. And, $C_{W-1}^R$ is the attention over temporal Jacobian in a time window $W - 1$ i.e. till previous step. In (Eq.10) we need score function which can estimate a denoised version of a noise-corrupted data sample given noise level $C(t)$. Comparatively, (Eq.13) also denoise the input noise accumulated over spatial Jacobians $C_W^P \to C(t)$ and the corrections applied to them $C_{W-1}^R \to$ rescaling $A(t)$ we are estimating the denoised Jacobian which is the correction term for spatial Jacobian. For convergence, as in (Eq.10) as soon as the corrections applied starts to contract it balance the opposing force from the error in spatial Jacobians and finally the corrections term reduced to zero. To ensure this correction term reduces to zero only for temporal noise we apply the flow matching (detailed balance) objective from GFlowNets, detailed subsequently.

### 3.4. Loss function

We distill the prior knowledge of a pretrained video diffusion model via **Video Score Distillation Sampling (SDS)**. Let $\mathbf{J} = \{\mathbf{J}_t\}_{t=0}^{N-1}$ (Eq.11) represent the predicted Jacobian fields over time, and $\mathbf{V}^* = \{\mathbf{V}_t^*\}_{t=0}^{N-1}$ be the temporally consistent set of vertices computed via differentiable function $g$ given $\mathbf{J}_t$ and $\mathbf{T}_c$ for each time step (Sec.2.1). For each time step $t$, the deformed mesh $\mathbf{M}_t^* = (\mathbf{V}_t^*, \mathbf{F})$ is used to warp $\mathcal{W}$ initial clipart image and then rendered using a differentiable renderer $\mathcal{R}$ (Li et al., 2020), producing frames $\mathbf{I}_t = \mathcal{R}(\mathcal{W}(I_0), \mathbf{M}_t^*)$. The sequence of frames forms the video $\mathbf{X} = \{\mathbf{I}_t\}_{t=0}^{N-1}$. The pretrained video diffusion model $\epsilon_\phi$ with the input video $\mathbf{X}$ as parameters, it produces a gradient w.r.t $\theta$ which are the spatial and temporal parameters driving the animation:

$$\nabla_\theta \mathcal{L}_{\text{SDS}}(\phi, \mathbf{X}) = \mathbb{E}_{t', \epsilon}\left[w(t')\left(\epsilon_\phi(\mathbf{z}_{t'}; \mathbf{y}, t') - \epsilon\right)\frac{\partial \mathbf{X}}{\partial \theta}\right], \tag{14}$$

where $t' \sim \mathcal{U}(0, T)$, $\epsilon \sim \mathcal{N}(0, \mathbf{I})$, and $\mathbf{z}_{t'}$ is a noisy latent embedding of $\mathbf{X}$, $\epsilon_\phi(\cdot)$ is the U-Net denoising network in the T2V model, conditioned on the text prompt $y$ and the diffusion time step $t'$. The optimization of SDS loss involves back-propagating through the UNet $\epsilon_\phi(\cdot)$, and then to spatial and temporal parameters, which is computationally expensive. Hence we leverage gradients as per (Poole et al., 2022) by omitting the UNet Jacobian. Specifically, in our

case gradient updates the Bézier parameters (spatial) and {attention, ODE} parameters (temporal) which refines the geometry of the meshes $\mathbf{M}^* = \{\mathbf{M}_t^*\}_{t=0}^{N-1}$. For better text-video alignment, classifier-free guidance (Ho & Salimans, 2022) is applied:

$$\epsilon_\phi(\mathbf{z}_{t'}; \mathbf{y}, t') \leftarrow (1+s)\epsilon_\phi(\mathbf{z}_{t'}; \mathbf{y}, t') - s\epsilon_\phi(\mathbf{z}_{t'}; \emptyset, t'), \quad (15)$$

where $s$ is the guidance scale, $\emptyset$ denotes null text prompt.

**Flow Matching Loss**   Optimizing the mesh deformation with Video SDS loss using Gaussian noise assumption can produce animations with abrupt transitions, exhibit local geometric distortions as the displacement for each keypoint is predicted independently and hence not be able to match the target distribution (as explained in Sec.2.2). In conclusion, we are required to optimize the temporal noise (Sec.3.3) for smooth evolution of keypoints and maintain local structure, however these pre-trained video diffusion models under isotropic Gaussian noise assumption increase the effective dimensionality of the data, which may begin as a low-dimensional manifold embedded within same dimensionality. Thus, maintaining intrinsic data dimensionality requires a choice of flow that preserves this dimension. We define the dimensionality as the denoising objective, for example, consider the noise quantified as $C(t)$

$$\mathbb{E}_{\mathbf{X}\sim\text{data}}\mathbb{E}_{n\sim\mathcal{N}(0,C(t))}\frac{\|D(\mathbf{X}+n; C(t)) - \mathbf{X}\|^2}{C(t)} \quad (16)$$

where $D(\cdot)$ de-noised version of a noise-corrupted data sample $\mathbf{X}$ given noise level $C(t)$. In practice we used the same denoiser $\epsilon_\phi$ U-Net of pre-trained video diffusion model, hence (Eq.16) is essentially the score function $\nabla_{\mathbf{X}} \log p(\mathbf{X})$.

To preserve the dimension for the temporal noise, despite learning the deformation under the Gaussian noise assumption, we propose to take the difference between the score function obtained for frames $\mathbf{I}_t$ produced through overall Jacobian $\mathbf{J}_t$ and spatial Jacobian $\mathbf{J}_t^P$. This loss is inspired from the detailed balance (DB) objective from GFlowNets where, the backward process (Spatial Jacobian) keeps on introducing the temporal noise and the forward process keeps on reducing the added temporal noise for each state transition. Since the score function is already normalized and comparable there is no normalizing factor $F(s)$ as in Sec.2.3, thus the loss function is given as:

$$L_{flow} = \mathbb{E}_{t',t}\|\nabla_{\mathbf{X}} \log p_{t'}(\mathbf{X}, \mathbf{J}_t) - \nabla_{\mathbf{X}} \log p_{t'}(\mathbf{X}, \mathbf{J}_t^P)\|^2 \quad (17)$$

Here, as well the gradient $\frac{\partial \mathbf{X}}{\partial \theta}$ is taken while back propagating from this loss. Moreover, we wanted the parameterized Bézier to capture temporal movement of keypoints as much as possible and hence we want the temporal Jacobian (corrective term) to be as small as possible, hence we add the

correction minimizing loss given as:

$$L_{flow} = \mathbb{E}_{t',t}\|\nabla_{\mathbf{X}} \log p_{t'}(\mathbf{X}, \mathbf{J}_t) - \nabla_{\mathbf{X}} \log p_{t'}(\mathbf{X}, \mathbf{J}_t^P)\|^2$$
$$+ \mathbb{E}_t\|\mathbf{J}_t - \mathbf{J}_t^P\|^2 \quad (18)$$

The **Overall Loss** for **FlexiClip** is defined as the weighted sum loss in Eq.14, 16, 17:

$$L_{SDS} + \lambda * L_{flow} \quad (19)$$

where $\lambda$ is the weight to balance the loss magnitudes. FlexiClip is end-to-end differentiable to be able to learn the keypoint movement and temporal smoothing required.

# 4. Experiments

We evaluate FlexiClip through comprehensive experiments. First, we describe the experimental setup (Sec.4.1) and evaluation metrics (Sec.4.2). Next, we compare FlexiClip with AniClipart (Wu et al., 2024), sketch-animation method Gal23 (Gal et al., 2024), as well as leading T2V models (Sec.4.3). Ablation studies (Sec.4.5) validate our design decisions, and we showcase FlexiClip's ability to handle complex animations (Sec.4.6).

## 4.1. Experimental Setup

FlexiClip enables high-resolution, complex animations by leveraging SVGs, where paths defined by control points are animated via mesh deformation and rendered into bitmaps using DiffVG (Li et al., 2020) for video SDS loss. For bitmap clipart, pixels within each triangle are warped. We used 30 clipart images from AniClipart (Wu et al., 2024) and additional ones from Freepik[1] across various categories (humans, animals, and objects), resized to 256×256 pixels. First, we learn $\mathbf{J}_0$ with Eq.4 for 10K iters. After that, motion trajectories with 8–11 control points were optimized upto 700 steps using Adam (learning rate: 0.5). We applied ModelScope T2V (Wang et al., 2023) with a guidance parameter of 50 for SDS loss. For spatial posing, cubic Bézier control points, we use a 4-layer MLP with LeakyReLU activation, with the final layer being linear. Temporal Jacobians from pfODE were predicted with a 3-layer MLP, while two attention networks with 32-dimensional keys/values and two heads modeled motion and deformation effectively. Standard 24-frame animations were rendered on an NVIDIA V100 in 40 minutes using 26 GB.

## 4.2. Metrics

We evaluated FlexiClip on same metrics from AniClipart (Wu et al., 2024), namely, 1) bitmap metrics (cosine similarity via CLIP for **visual identity** and X-CLIP for **text-video alignment**) and 2) animation metrics (Motion Vibrancy

---

[1]https://www.freepik.com/

*Table 1.* Quantitative results of FlexiClip in terms of bitmap metrics against AniClipart and T2V/I2V models.

| Method | Visual Identity Preservation (CLIP Score ↑) | Text-Video Alignment (X-CLIP Score ↑) |
|---|---|---|
| DynamiCrafter (Xing et al., 2025) | 0.8031 | 0.1732 |
| Gal23 (Gal et al., 2024) | 0.8395 | 0.1865 |
| VideoCrafter2 (Chen et al., 2024) | 0.8410 | 0.1988 |
| I2VGen-XL (Zhang et al., 2023) | 0.8798 | 0.2015 |
| ModelScope (Wang et al., 2023) | 0.8632 | 0.2037 |
| ToonCrafter (Xing et al., 2024) | 0.9280 | 0.1997 |
| AnimateLCM-I2V (Wang et al., 2024) | 0.9274 | 0.2020 |
| Pyramid Flow (Lei et al., 2023) | 0.9312 | 0.2045 |
| LTXVideo (HaCohen et al., 2024) | 0.9325 | 0.2054 |
| AniClipart (Wu et al., 2024) | 0.9401 | 0.2075 |
| **FlexiClip** (Ours) | 0.9563 | 0.2102 |

*Table 2.* Quantitative results of FlexiClip in terms of animation metrics against AniClipart.

| Method | MV↑ | TC↓ | GD↓ | DS↓ | AE ($\times 10^3$) ↑ |
|---|---|---|---|---|---|
| AniClipart | 20.87 | 8.51 | 50.98 | 18.49 | 75.23 |
| FlexiClip (Ours) | 25.33 | 8.14 | 52.34 | 13.76 | 113.44 |

(**MV**), Temporal Consistency (**TC**), and Geometric Deviation (**GD**)). To better evaluate methods for free-flowing and smooth animation we propose the following metrics:

**Deformation Smoothness(DS):** This metric evaluates how smoothly the control points deform along Bézier paths. It computes the average difference in displacement vectors of consecutive frames:

$$DS = \frac{1}{(N-1)*M} \sum_{t=1}^{N-1} \sum_{i=0}^{M-1} \|d_t^{(i)} - d_{t-1}^{(i)}\|,$$

where $d_t^{(i)}$ is the displacement of control point $i$ at frame $t$. Lower values indicate smoother deformation.

**Animation Energy(AE):** To evaluate the energy distributed across control points, we calculate the mean squared displacement across all frames:

$$AE = \frac{1}{N \cdot M} \sum_{t=1}^{N} \sum_{i=0}^{M-1} \|p_t^{(i)} - p_0^{(i)}\|^2.$$

Higher values indicate more dynamic animations.

### 4.3. Comparison to State-of-the-Art Methods

**FlexiClip versus AniClipart:** We compare FlexiClip with AniClipart based on the results observed in the tables (Tab.1, 2 & 3) and visual examples (Fig.3). One of the key differences between the two models lies in their ability to preserve visual identity and create more coherent text-video animations. While AniClipart demonstrates impressive performance in visual identity preservation and text-video alignment, it encounters challenges when it comes to handling complex deformations. For instance (Fig.3), in the case of

hand deformation (e.g., during dance movements), AniClipart exhibits noticeable distortions, which negatively impact the naturalness of the animation. FlexiClip, on the other hand, maintains better consistency in object shape and deformation, showing smoother transitions without compromising visual identity. In particular, when handling leg folding during jumping and dynamic hand movements, FlexiClip ensures these actions are portrayed realistically (whereas AniClipart shows the distorted hand for girl/boy jumping), which is a critical requirement for creating natural animations.

Additionally, FlexiClip excels at producing smoother and more realistic deformations, as evidenced by the parrot animation, where the wing flapping looks much more natural and consistent. The wings of the parrot exhibit a smooth, continuous motion, enhancing the visual realism compared to AniClipart's deformation, which is flapping its tail rather than wings showing lack of text alignment. Looking at the quantitative results (Tab.1 and 2), FlexiClip outperforms AniClipart in both the bitmap and animation metrics. Specifically, FlexiClip achieves a higher CLIP score (0.9563 vs. 0.9401) and X-CLIP score (0.2102 vs. 0.2075), signifying better visual identity preservation and text-video alignment. Furthermore, FlexiClip's animation quality metrics, including MV, TC, GD, DS, and AE, also surpass AniClipart's results, with FlexiClip yielding more motion variation, lower distortions, and higher animation efficacy. Notably FlexiClip GD has increased but DS decreased showing smoother animations. **FlexiClip versus I2V Models:** See App.B.

**User Study** To assess the improvements made by FlexiClip, we conducted a subjective user study with 55 static clipart images animated by six methods: FlexiClip (Ours), AniClipart, LTXVideo, PyramidFlow, AnimateLCM-I2V, and DynamiCrafter. Participants were asked to rate the animations on visual identity preservation, text-video alignment, and smoothness using a six-point scale from 0 (strongly disagree) to 1.0 (strongly agree). The study was conducted online with 30 participants, and ratings were averaged across the 55 clipart images. The results in Tab.3 show that FlexiClip significantly outperforms the other methods.

*Table 3.* Subjective user study results

| | User Selection% ↑ | | |
|---|---|---|---|
| | Identity Preservation | Text-Video Alignment | Smoothness |
| **FlexiClip (Ours)** | **94.90** | **94.54** | **93.81** |
| AniClipart | 83.63 | 80.72 | 76.36 |
| LTXVideo | 61.82 | 60.36 | 58.18 |
| PyramidFlow | 56.36 | 54.90 | 52.36 |
| AnimateLCM-I2V | 49.09 | 48.72 | 45.09 |
| DynamiCrafter | 2.18 | 1.82 | 1.09 |

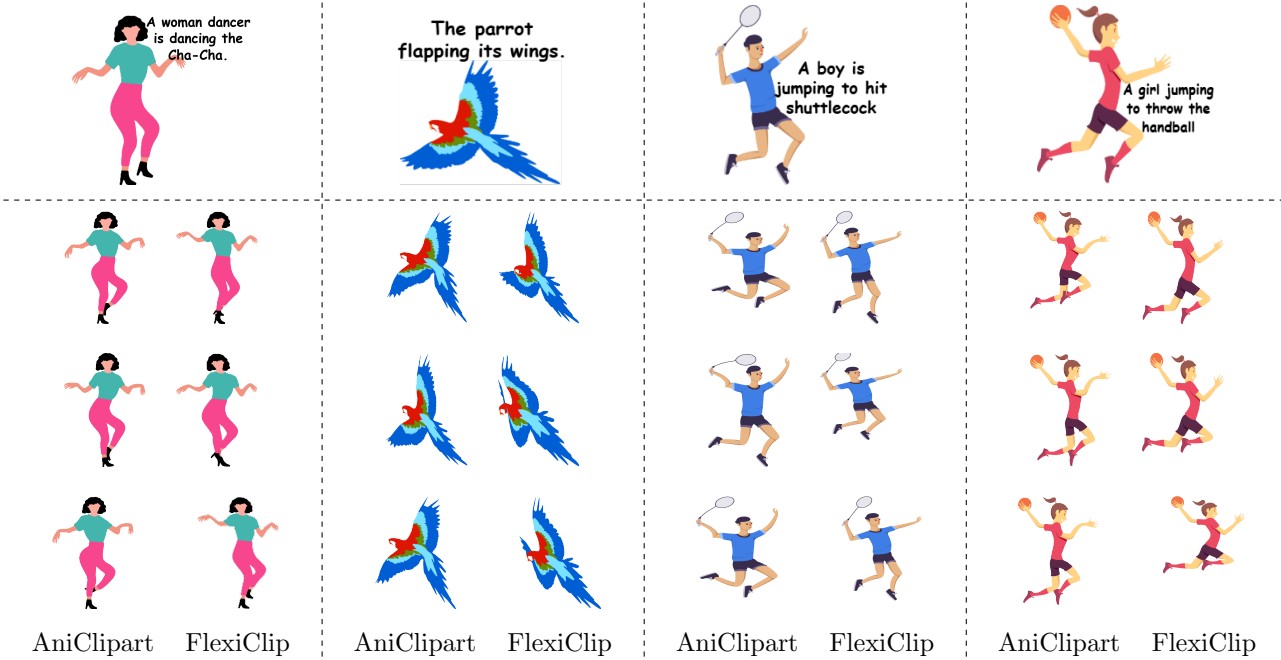

*Figure 3.* **FlexiClip vs AniClipart**: Four consecutive frames are shown for comparison. AniClipart distorts the objects (e.g., boy/girl jumping, woman dancing), lacks proper text conditioning (e.g., parrot), and shows poor temporal consistency (e.g., boy jumping). In contrast, FlexiClip preserves the visual identity, maintaining consistent shapes and producing high-quality, smooth, and text-aligned animations. Detailed animations can be seen at: https://creative-gen.github.io/flexiclip.github.io/

### 4.4. Effect of $\lambda$ on Motion Quality and Convergence

In our experiments, we found that the $\lambda$ parameter plays a critical role in balancing the alignment of generated motion with the text prompt and ensuring smooth, natural motion trajectories. Specifically, tuning $\lambda$ influences both the quality of the animation and the convergence behavior during training. We set $\lambda = 15$ for all reported results to maintain consistency across examples.

When $\lambda$ is set too low, such as around 1, the model struggles to generate coherent motion aligned with the textual description, often requiring up to 1000 gradient descent steps to produce acceptable outputs. Increasing $\lambda$ incrementally to values up to 5 does not result in a significant reduction in the required number of training steps, indicating limited benefit in that range. However, a more noticeable impact emerges when $\lambda$ is raised to values between 5 and 10, where we observe variability in convergence, some animations achieve satisfactory quality in as few as 600 steps, while others still require up to 900 steps.

At $\lambda = 15$, the motion generation becomes consistently more reliable, with most animations aligning well with the text prompt after approximately 700 training steps. This value represents a practical trade-off between convergence speed and motion quality. Pushing $\lambda$ beyond 15 leads to noticeably sharper and faster motions, but this comes at the expense of temporal smoothness, resulting in less natural-

looking animations. Thus, $\lambda = 15$ serves as an effective default for balancing alignment, smoothness, and training efficiency.

### 4.5. Ablation Study

We conducted an ablation study to validate the significance of each critical component in FlexiClip. The quantitative results of the study are detailed in Table 4, while Figure 6 and 7 showcases a qualitative comparison of the different variants. We performed ablation to describe importance of temporal Jacobian obtained from pfODE and the flow matching loss, which constitute the core of our system. Notably, the geometric deviation (GD) in FlexiClip is higher compared to AniClipart due to the absence of ARAP deformation, which inherently minimizes shape distortion observed from reduced deformation smoothness(DS).

**Temporal Jacobian**    To evaluate the effectiveness of incorporating temporal Jacobian, we removed this component and instead used a baseline keypoint transformation without considering the continuous dynamics provided by the pfODE framework. Without temporal Jacobian, the animation exhibited noticeable artifacts (Fig.6), including a lack of smoothness in movements and unnatural keypoint transitions, as seen in the parrot, ghost, and cloud examples in (Fig.6). The quantitative results corroborate these findings: the "w/o Temporal Jacobian" variant exhibited a

lower motion variance (MV = 23.00) and higher temporal inconsistency (TC = 8.80). Additionally, the geometric distortion (GD) score for this variant decreased to 51.50 (vs Default) showing rigid transformation as seen from ghost, parrot example it failed to show movement of hands and wings flapping respectively. In contrast, the default setting, with temporal Jacobians, yielded superior results across all metrics, highlighting its necessity for realistic and accurate animations.

*Table 4.* Ablation Study with FlexiClip variants.

| FlexiClip Variant | MV ↑ | TC ↓ | GD ↓ | DS ↓ | AE ↑ |
|---|---|---|---|---|---|
| w/o Temporal Jacobian | 23.00 | 8.80 | 51.50 | 14.00 | 105.00 |
| w/o Flow Match. Loss | 24.50 | 8.40 | 53.00 | 14.20 | 95.00 |
| Default | 25.33 | 8.14 | 52.34 | 13.76 | 113.44 |

**Flow Matching Loss**    The flow matching loss plays a crucial role in aligning the generated motion trajectories with the temporal coherence and dynamics inferred from text descriptions. To assess its impact, we omitted the flow matching loss from the total loss. As shown in the man and woman dance examples in Fig.7, this substitution led to erratic and exaggerated movements, deviating significantly from the intended motion semantics. Quantitatively, "w/o Flow Match. Loss" variant showed increased geometric distortion (GD = 53.00) and reduced average energy (AE = 95.00), indicative of less expressive and abrupt motion dynamics. Additionally, temporal consistency (TC) degraded slightly (TC = 8.40), reinforcing the importance of this loss in maintaining stable frame-to-frame transitions. The absence of flow matching loss caused inconsistencies visible in the chaotic limb movements and unstable/abrupt posture transitions in the dancing examples. The default setting, which incorporates flow matching loss, demonstrated a balanced trade-off between dynamism and coherence, achieving the highest average energy (AE = 113.44) and a competitive motion variance (MV = 25.33) with smooth animation.

### 4.6. More Results

FlexiClip has been able to support diverse and complex animations (Tab.5), broadening its applications in creating visually captivating and textually coherent animations.

**Rotation**    Demonstrated rotational movements. The flower swaying its petals in the breeze showcases FlexiClip's ability to handle smooth rotational dynamics. Without any rotational mechanism explicitly, the system captures the gentle oscillation of petals, creating a lifelike representation of natural motion.

**Multiple Text Conditions**    Demonstrated complex motions via multiple text conditioning. The woman in a green dress with black polka dots demonstrates FlexiClip's capa-

bility to animate based on complex textual conditions. Here, she dances and folds her hands in rhythm with the description, reflecting the system's ability to integrate fine-tuned motion semantics with precise keypoint dynamics.

**Multiple Objects**    Demonstrated coordinated interactions. The couple dancing exemplifies FlexiClip's support for animating multiple objects simultaneously. The synchronized movements of the man and woman highlight the system's proficiency in managing interactions between objects while maintaining spatial and temporal coherence.

**Layered Animations**    FlexiClip supports layered animations (see in Fig.1 Breakdancer animation) for depth and complexity. This allows for the creation of dynamic animations enhancing its applicability for interactive media.

*Table 5.* FlexiClip: Diverse and Complex Animations.

## 5. Conclusion

We introduced FlexiClip, a cutting-edge framework for text-to-animation generation, excelling in visual identity preservation, text-video alignment, and animation smoothness. Extensive evaluations, including user studies, demonstrate that FlexiClip outperforms existing methods like AniClipart and LTXVideo, particularly in handling complex deformations, maintaining temporal consistency, and producing realistic motions. Ablation studies validated the importance of temporal Jacobians and flow matching loss, essential for smooth transitions and accurate motion dynamics. FlexiClip supports diverse scenarios, including rotational dynamics, multi-object interactions, and layered animations, making it versatile for interactive media and entertainment. By setting a new standard in text-to-animation frameworks, FlexiClip paves the way for future advancements in real-time applications, 3D integration, and personalized animations.

## Impact Statement

This paper introduces **FlexiClip**, a novel framework for animating static clipart images with enhanced temporal coherence and geometric integrity. By integrating temporal Jacobians, continuous-time modeling via probability flow ODEs (pfODEs), and a flow matching loss inspired by GFlowNet principles, FlexiClip addresses longstanding challenges in clipart animation, such as abrupt motions and geometric distortions.

### Potential Positive Impacts

- **Advancement in Digital Animation:** FlexiClip's methodology can significantly benefit industries involved in digital animation, including education, entertainment, and digital marketing, by enabling the creation of smooth and natural animations from static images.

- **Accessibility for Content Creators:** By simplifying the animation process for clipart images, FlexiClip can empower individual creators and small businesses to produce high-quality animations without extensive resources.

- **Educational Tools Enhancement:** The ability to animate educational clipart can lead to more engaging learning materials, potentially improving educational outcomes.

### Potential Negative Impacts and Mitigations

- **Misinformation Risks:** The ease of animating static images could be misused to create deceptive content. To mitigate this, it's essential to develop and integrate detection tools that can identify AI-generated animations.

- **Intellectual Property Concerns:** Animating copyrighted clipart without proper authorization could infringe on intellectual property rights. Users should be educated on copyright laws, and systems should include checks to prevent unauthorized use.

- **Bias in Animation Outputs:** If the training data for FlexiClip contains biased representations, the animations produced might perpetuate these biases. It's crucial to use diverse and representative datasets and implement bias detection mechanisms.

### Ethical Considerations

While FlexiClip primarily aims to advance the field of machine learning and animation, it is important to remain vigilant about its applications. Developers and users should adhere to ethical guidelines, ensuring that the technology is used responsibly and does not contribute to societal harm.

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

# A. Ablation Studies

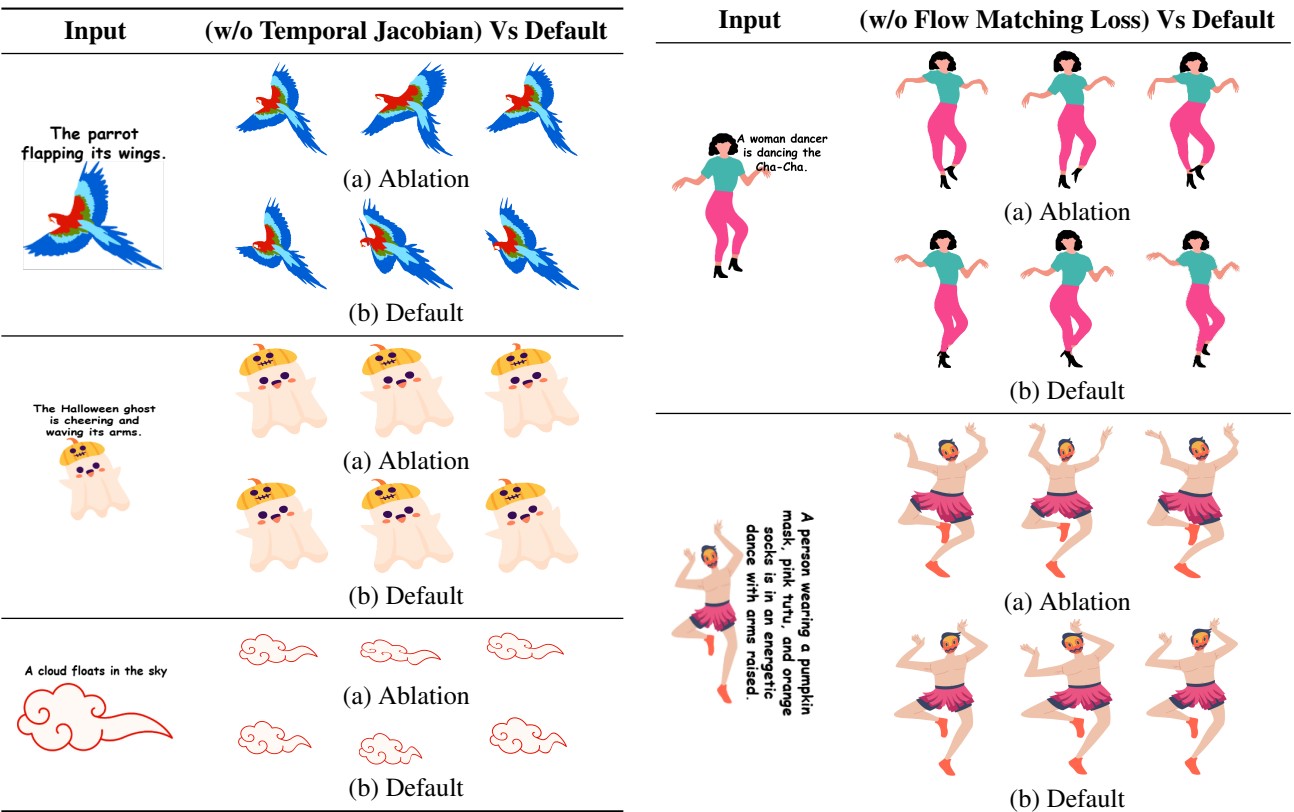

*Table 6.* Ablation: w/o Temporal Jacobian vs Default- This figure compares animations with and without the temporal Jacobian. Without it, noticeable artifacts such as jerky movements and unnatural keypoint transitions appear (e.g., parrot, ghost, cloud). In contrast, the temporal Jacobian yields smoother, more consistent, and realistic animations.

*Table 7.* Ablation: w/o Flow Matching Loss vs Default - Removing the flow matching loss leads to erratic, exaggerated motion with chaotic limb movements and unstable transitions. Including the flow matching loss preserves smooth, dynamic, and coherent animations, ensuring stable frame-to-frame transitions

**w/o Temporal Jacobian:** To evaluate the effectiveness of incorporating temporal Jacobian, we removed this component and instead used a baseline keypoint transformation without considering the continuous dynamics provided by the pfODE framework. Without temporal Jacobian, the animation exhibited noticeable artifacts (Fig.6), including a lack of smoothness in movements and unnatural keypoint transitions, as seen in the parrot, ghost, and cloud examples in Fig.6. The quantitative results corroborate these findings: the "w/o Temporal Jacobian" variant exhibited a lower motion variance (MV = 23.00) and higher temporal inconsistency (TC = 8.80). Additionally, the geometric distortion (GD) score for this variant decreased to 51.50 (vs Default) showing rigid transformation as seen from ghost, parrot example it failed to show movement of hands and wings flapping respectively. In contrast, the default setting, with temporal Jacobians, yielded superior results across all metrics, highlighting its necessity for realistic and accurate animations.

**w/o Flow Matching Loss:** The flow matching loss plays a crucial role in aligning the generated motion trajectories with the temporal coherence and dynamics inferred from text descriptions. To assess its impact, we omitted the flow matching loss from the total loss. As shown in the man and woman dance examples in Fig.7, this substitution led to erratic and exaggerated movements, deviating significantly from the intended motion semantics. Quantitatively, the "w/o Flow Match. Loss" variant showed increased geometric distortion (GD = 53.00) and reduced average energy (AE = 95.00), indicative of less expressive and abrupt motion dynamics. Additionally, temporal consistency (TC) degraded slightly (TC = 8.40), reinforcing the importance of this loss in maintaining stable frame-to-frame transitions. The absence of flow matching loss caused inconsistencies visible in the chaotic limb movements and unstable/abrupt posture transitions in the dancing examples. The default setting, which incorporates flow matching loss, demonstrated a balanced trade-off between dynamism and coherence, achieving the highest average energy (AE = 113.44) and a competitive motion variance (MV = 25.33) with

smooth animation.

## B. FlexiClip Vs T2V/I2V Models

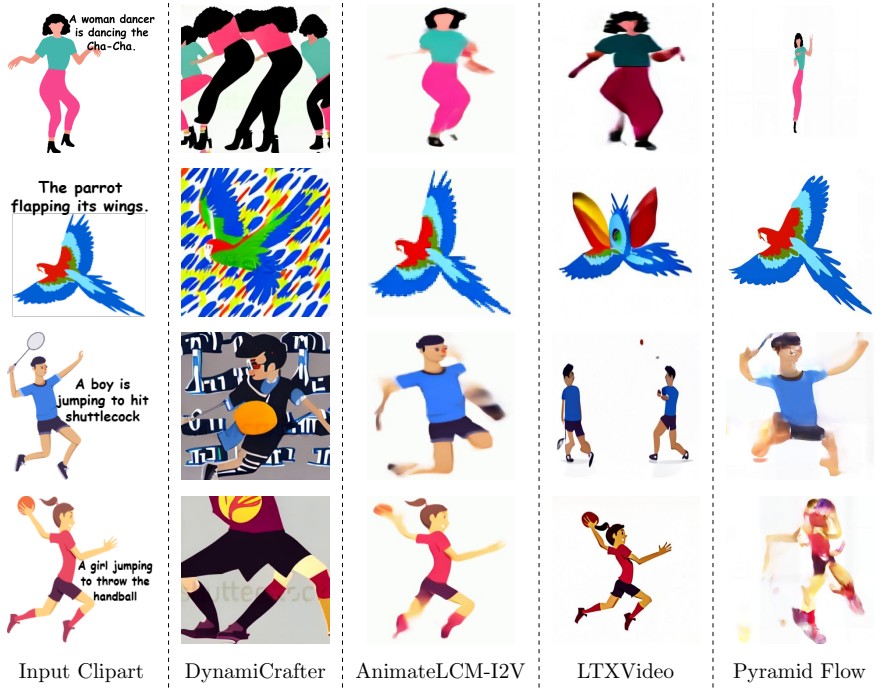

| Input Clipart | DynamiCrafter | AnimateLCM-I2V | LTXVideo | Pyramid Flow |

*Figure 4.* **FlexiClip vs T2V/I2V Models**: LTXVideo and PyramidFlow show moderate performance, with scores higher than methods like DynamiCrafter and AnimateLCM-I2V, but still lagging behind FlexiClip. While these methods manage basic identity preservation, they struggle with animation movement and text alignment, resulting in lower ratings. DynamiCrafter, in particular, performs poorly across all metrics, reflecting its inability to preserve visual details and adapt effectively to text prompts.

The methods evaluated in this study exhibit varying performance levels when it comes to key animation quality aspects, such as identity preservation, text-video alignment, and smoothness. While certain techniques are successful in preserving the overall semantics of the input clipart, they often fall short in maintaining finer details, leading to lower identity preservation scores. These methods tend to produce animations that are static or lack significant motion, which negatively impacts the alignment with the given text prompts, ultimately resulting in less engaging and less accurate animations. Other methods, such as AniClipart and LTXVideo, perform relatively well but still struggle with capturing finer details and translating text prompts into dynamic animations with sufficient movement. These approaches yield more static animations, which limits their ability to accurately reflect the intended transformations and reduces the overall visual appeal.

On the other hand, methods like DynamiCrafter exhibit the weakest performance in all areas, with noticeably poor identity preservation and minimal alignment with text descriptions. The animations generated by these techniques tend to lack both the required detail and movement, making them less suitable for generating high-quality, contextually accurate animations.

Overall, the comparison highlights the significant improvements offered by FlexiClip over other methods, demonstrating its ability to generate high-quality, detailed, and smooth animations that align more effectively with input prompts.

## C. Derivation of the Fokker-Planck Equation Under time varying densities

We start with the smoothing kernel defined as:

$$\kappa(x,t) \equiv N(x;\mu,C(t)) = \frac{1}{(2\pi)^{d/2}\det(C)^{1/2}} \exp\left(-\frac{1}{2}(x-\mu)^\top C^{-1}(x-\mu)\right),$$

where:

- $\mu$ is the mean,

- $C(t)$ is the covariance matrix,

- $x \in \mathbb{R}^d$.

## C.1. Derivatives of the Kernel

### C.1.1. TIME DERIVATIVE

The time derivative of $\kappa(x, t)$ is given by:

$$\frac{\partial \kappa}{\partial t} = \kappa(x, t) \left[ -\frac{1}{2} \operatorname{tr}(C^{-1}\dot{C}) + \frac{1}{2} \operatorname{tr}\left( C^{-1}(x - \mu)(x - \mu)^\top C^{-1}\dot{C} \right) \right].$$

### C.1.2. GRADIENT

The gradient of $\kappa(x, t)$ is:

$$\nabla \kappa = -C^{-1}(x - \mu)\kappa.$$

### C.1.3. SECOND DERIVATIVE

The second derivative is:

$$\frac{\partial^2 \kappa}{\partial x_i \partial x_j} = \kappa \left[ (C^{-1}(x - \mu))_i (C^{-1}(x - \mu))_j - (C^{-1})_{ij} \right].$$

## C.2. Evolution of the Probability Density

The time varying probability density $p(x, t)$ evolves as:

$$p(x, t) = p_0(x) * \kappa(x, t),$$

where $*$ denotes convolution. The evolution is governed by:

### C.2.1. TIME DERIVATIVE OF $p(x, t)$

$$\frac{\partial p}{\partial t} = p_0(x) * \frac{\partial \kappa}{\partial t}.$$

Substituting the time derivative of $\kappa$, we have:

$$\frac{\partial p}{\partial t} = p_0(x) * \kappa \left[ -\frac{1}{2} \operatorname{tr}(C^{-1}\dot{C}) + \frac{1}{2} \operatorname{tr}\left( C^{-1}(x - \mu)(x - \mu)^\top C^{-1}\dot{C} \right) \right].$$

### C.2.2. DIVERGENCE OF THE DRIFT TERM

For a drift vector field $f(x)$, the divergence term is:

$$-\nabla \cdot (fp) = -p_0(x) * \nabla \cdot (f\kappa),$$

where:

$$\nabla \cdot (f\kappa) = (\nabla \cdot f)\kappa - f \cdot \nabla \kappa.$$

### C.2.3. DIFFUSION TERM

The diffusion term, assuming the diffusion matrix $G(x, t)$, becomes:

$$\frac{1}{2}\nabla_i \nabla_j \left[ \sum_k G_{ik} G_{jk} p \right] = \frac{1}{2} p_0(x) * \kappa \sum_{ij} \left[ \nabla_i \nabla_j \sum_k G_{ik} G_{jk} \right].$$

### C.3. Simplifications and Final Form

Assume $f(x) = 0$ (no drift) and $\nabla_i G_{jk} = 0$ (spatially homogeneous diffusion). The condition simplifies to:

$$-\frac{1}{2}\operatorname{tr}(C^{-1}\dot{C}) + \frac{1}{2}\operatorname{tr}\left(C^{-1}(x-\mu)(x-\mu)^\top C^{-1}\dot{C}\right) = -\frac{1}{2}\operatorname{tr}(C^{-1}GG^\top) + \frac{1}{2}\operatorname{tr}\left(C^{-1}(x-\mu)(x-\mu)^\top C^{-1}GG^\top\right).$$

This condition is satisfied when:

$$GG^\top(x,t) = \dot{C}(t).$$

