# OpenReview forum: "FlexiClip: Locality-Preserving Free-Form Character Animation"
_ICML.cc/2025/Conference — ICML 2025 poster_

### Official Review · Reviewer_3qmH · 2025-03-12

**Overall Recommendation:** 3

**Summary:**

In this paper, the authors propose a new method, named as FlexiClip, to achieve better temporal coherence and geometric consistency in animated clipart. To better preserve motion smoothness without introducing geometric distortions, FlexiClip utilizes Probability flow ODE (pfODE) to model the evolution of temporal Jacobian which incrementally adjust the spatial geometry over time to account for temporal variations. As a key technical contribution, a flow matching loss is further used to optimize the reduction of temporal noise. Qualitative and quantitative comparisons are performed to validate the effectiveness of the proposed method.

**Claims And Evidence:**

Please see the comments below.

**Essential References Not Discussed:**

I think the current related work part is okay.

**Experimental Designs Or Analyses:**

Yes, please see the **Claims And Evidence** part.

**Methods And Evaluation Criteria:**

Yes. I think the proposed method is technically sound.

**Other Comments Or Suggestions:**

No.

**Other Strengths And Weaknesses:**

**Strengths**:
1. This paper is well written. I can follow it easily.
2. From my understanding, I like the basic idea of FlexiClip which is simple and clear.
3. The authors perform sufficient experiments to verify the effectiveness of the proposed method.

**Weakness**:

I am not an expert in this area so I just have some ideas to help authors refine this work.
1. The visual improvement over AniClipart is limited. For example, in Figure 3, both AniClipart and FlexiClip can synthesize good shapes with adaptive details. I also cannot perceive meaningful differences between these two methods in the supplementary website.
2. The current experimental results can only support rigid motion rather than non-rigid complicated motions. Could you please provide more challenging demo results in the revision?
3. Can the key contribution of FlexiClip be extended to 3D shapes? Can the authors share some insights for it?
4. Could you also conduct some user studies for the ablation study? For some examples shown in Table 6-7, the visual differences between the ablated model and the full model are minor. Adding some user study results could help to achieve comprehensive conclusions.
5. Typos: L22-L23: "Additionally, Recent" -> "Additionally, recent";

---
## Update After Rebuttal

I thank the authors for providing such detailed response which address most of my concerns. Thus I will remain a positive tone towards this submission.

**Questions For Authors:**

No.

**Relation To Broader Scientific Literature:**

I think this submission has already listed sufficient related papers.

**Theoretical Claims:**

I have checked the proofs in Section C.

---

> ### Author Rebuttal · Authors · 2025-03-26
>
> Thank you for your thoughtful review. Below, we provide detailed responses to each of the weaknesses you raised:
>
> 1. **Figure 3 Analysis**: Upon closer inspection, you can observe that AniClipart distorts objects (e.g., hand distortion in the boy/girl jumping and woman dancing examples), lacks proper text conditioning (e.g., the parrot), and exhibits poor temporal consistency (e.g., the boy jumping). If the figures are unclear, we encourage you to view the detailed versions on our anonymous webpage: https://creative-gen.github.io/flexiclip.github.io/.
>
> 2. **Additional Results**: The webpage contains more extensive results, showcasing a variety of motion complexities, including couple dancing and two-object interactions. We are also in the process of adding more results, including 3D motion animations.
>
> 3. **3D Extension**: Thank you for your question regarding 3D extensions. FlexiClip can be easily adapted for 3D animation, as it only requires adding one extra coordinate and learning the Bézier trajectory over it. This modification is straightforward for our approach. We will release the code with configurable 2D and 3D settings.
>
> 4. **User Studies on Ablation Results**: We will conduct user studies on the ablated results as well. However, if you are referring to minor differences between the ablated and full models, we recommend checking the videos on our webpage. While the images may not clearly highlight these differences, the video outputs of the full and ablated models exhibit significant variations. We will include these results in the final draft for further user study.
>
> 5. **Typographical Errors**: We appreciate your feedback regarding typos and will carefully proofread the final revision to ensure all errors are corrected.
>
> Once again, we sincerely appreciate your time and effort in reviewing our paper. We kindly request you to consider our responses and, if appropriate, please increase your score accordingly.

---

> > ### Comment · Reviewer_3qmH · 2025-04-03
> >
> > Dear Authors,
> >
> > I thank you for your feedback and trying to address my concerns. While the last three points have been addressed, I still am worried about the first two points. Probably it is somewhat subjective but I still cannot perceive big differences in terms of visual comparisons. Also, the presented motions are also a bit simple. As I am not an expert in this area, I would make the decision upon other reviewers' comments.

---

> > > ### Author Response · Authors · 2025-04-03
> > >
> > > Thanks for letting me know this. Here I am detailing more on the first two points:
> > >
> > > **Your first point regarding Figure 3**
> > >
> > > In Figure 3 there are four animations lets go over each of them one by one: (I request again to see all these animations on the GitHub page:  https://creative-gen.github.io/flexiclip.github.io/.)
> > >
> > > 1. In the first animation there is a woman dancing if you compare both methods in AniClipart generates animation where hand of girl is distorted while it is not the case with our proposed solution FlexiClip.
> > >
> > > 2. In the second animation we are asking in the prompt to flap the wings of the bird, but if you see the animation generated by AniClipart it is only flapping the tail not the wings hence there is poor text alignment there, while the animation generated by FlexiClip is actually flapping the wings this is the clear difference between animations generated by AniClipart and FlexiClip.
> > >
> > > 3. In the third animation we wanted the boy to jump and hit the shuttlecock and in order to do that the requirement is there must be no (absolute zero) distortion in any of its body part. But if you check the animations generated by AniClipart it is distorting the hands of the boy while FlexiClip is not at all distorting any its body parts. See the actual motion on the page as well as the FlexiClip motion is much smoother than the one generated by AniClipart.
> > >
> > > 4. In the fourth animation, we wanted the girl to throw the handball while jumping, but again here the animation generated by AniClipart is distorting the hands of the girl while FlexiClip is generating the zero distortion animation and if you check the actual animation in GitHub page the jump is much more smoother rather than quick and abrupt jump in AniClipart.
> > >
> > > **Your Second Point regarding simple animations**
> > >
> > > I again request you to please watch the GitHub page: https://creative-gen.github.io/flexiclip.github.io/
> > >
> > > 1. There are four animations in which two persons (couple, multi-object) animation is shown to animate in a single go. This is not simple since generating the multi-object animation requires modelling two persons at the same time and make there motion coherent to each other with just a simple SDS loss is a fairly complex task.
> > >
> > > 2. There are other animations like girl swaying on the hammock is also another example of complex task since the hammock and girl movement are generated in sync with each other.
> > >
> > > 3. Additionally, we showed the multi-condition animation as well where we show the single animation to follow multiple conditions specified in the prompt.
> > >
> > > 4. The swaying example of flower with rotation shows the additional level of complexity we have introduced.
> > >
> > > Request you to please try to go over the GitHub page: https://creative-gen.github.io/flexiclip.github.io/
> > >
> > > Again, we value your time and energy it requires to review our paper. Many thanks for reviewing our paper.

---

### Official Review · Reviewer_LSdo · 2025-03-13

**Overall Recommendation:** 3

**Summary:**

The paper proposes FlexiClip,  a novel approach designed to overcome these limitations by addressing the intertwined challenges of temporal consistency and geometric integrity, which extends traditional Bezier curve-based trajectory modeling with (1) temporal Jacobians to correct motion dynamics incrementally, (2) continuous-time modeling via pfODEs to mitigate temporal noise, and (3) a flow matching loss inspired by GFlowNet principles to optimize smooth motion transitions. The method sets a new standard for clipart animation.

**Claims And Evidence:**

Yes. The paper's core claims about FlexiClip are well-supported by compelling evidence. Visual identity preservation is validated through higher CLIP scores (0.9563 vs 0.9401) and 94.90% user preference.Temporal consistency improvements are demonstrated via lower TC scores (8.14 vs 8.51) and effective ablation studies. Text-video alignment claims are substantiated by improved X-CLIP scores (0.2102 vs 0.2075) and 94.54% user preference in alignment ratings.

**Essential References Not Discussed:**

N/A

**Experimental Designs Or Analyses:**

Yes, the comparison and ablation study are efficient. The sample size (30 participants) for the user study is moderate but acceptable.

**Methods And Evaluation Criteria:**

Yes, the FlexiClip method makes sense to me. The bitmap metrics, animation metrics used and the proposed deformation smoothness and animation energy also make sense to me.

**Other Comments Or Suggestions:**

N/A

**Other Strengths And Weaknesses:**

Other strengths:
The development of novel metrics (DS and AE) shows thoughtfulness in how animation quality should be measured.
The paper clearly identifies and addresses a specific limitation in prior work, the trade-off between spatial coherence and temporal consistency in clipart animation.

Other weaknesses:
The evaluation focuses on relatively simple character animations.

**Questions For Authors:**

How sensitive is the method to the choice of hyperparameters, particularly the λ weight balancing the SDS loss and flow matching loss?

**Relation To Broader Scientific Literature:**

The FlexiClip paper makes three key innovative contributions to animation research: (1) It introduces temporal Jacobians that incrementally correct spatial deformations, addressing the frame-to-frame inconsistency problems in prior works like AniClipart (Wu et al., 2024); (2) It adapts probability flow ODEs (originally from generative modeling by Song et al., 2020) to animation contexts, creating a continuous-time framework that preserves geometric structure during complex movements; and (3) It implements a flow matching loss inspired by GFlowNet principles (Bengio et al., 2023) that optimizes smooth transitions between frames.The significance lies in FlexiClip's mathematical formulation that unifies spatial and temporal modeling, effectively solving long-standing challenges in clipart animation where visual identity preservation must be balanced with natural motion dynamics.

**Theoretical Claims:**

Yes. The main theoretical foundation in Section 2.2 and Appendix C, where the authors develop their continuous-time modeling approach using probability flow ODEs. I didn't find any obvious errors in their mathematical derivations.

---

> ### Author Rebuttal · Authors · 2025-03-31
>
> **Sensitivity of λ in Balancing SDS Loss and Flow Matching Loss**
>
> Thank you for your thoughtful review and for raising this important question regarding the sensitivity of our method to the hyperparameter $\lambda$, which balances the SDS loss and flow matching loss.
>
> Through our experiments, we observed that tuning $\lambda$ is crucial for balancing trajectory alignment with text and ensuring smooth motion. In our setup, we set $\lambda$ = 15 for consistency across all examples.
>
> 1. When $\lambda=1$, the model requires significantly more gradient updates—often up to 1000 steps—to generate acceptable motion.
> 2. Increasing $\lambda$ up to 5 does not significantly reduce the number of required steps.
> 3. Raising $\lambda$ from 5 to 10 introduces some variability, where good results can sometimes be achieved in 600 steps, but in other cases, up to 900 steps are needed.
> 4. At $\lambda=15$, nearly all generated motions align well with the text prompt at around 700 steps.
> 5. Further increasing $\lambda$ results in sharper and faster motion, which appears less smooth.
>
> We appreciate this insightful question and will use the additional page allowance in the final paper to provide further details and examples.
>
> Additionally, we invite you to explore our GitHub page (https://creative-gen.github.io/flexiclip.github.io/), where we showcase more complex motion examples beyond those currently visible in our paper which are deemed to be simpler animations.
>
> We sincerely appreciate your valuable feedback. We kindly request you to consider our responses and, if appropriate, please increase your score accordingly.

---

> > ### Comment · Reviewer_LSdo · 2025-04-07
> >
> > Thank you for your detailed explanation on λ sensitivity. I look forward to your revised version with additional details on hyperparameter tuning. Additionally, after reviewing your GitHub page and observing the dynamic results, I am convinced of the effectiveness of your approach. The complex motion examples are impressive and clearly demonstrate significant improvement compared to existing work like DynamiCrafter and LTXvideo. I would like to upgrade my recommendation to Accept for your work.

---

> > > ### Author Response · Authors · 2025-04-07
> > >
> > > Thank you so much, and again many thanks for reviewing our paper, we appreciate the time and effort you put in to review the paper carefully.

---

### Official Review · Reviewer_Ry9s · 2025-03-13

**Overall Recommendation:** 3

**Summary:**

The paper proposes FlexiClip, a novel method for animating clipart images while preserving temporal coherence and geometric integrity. It extends existing approaches by incorporating temporal Jacobians for incremental motion correction, probability flow ODEs (pfODEs) for continuous-time modeling, and flow matching loss inspired by GFlowNet principles. The paper claims that these enhancements lead to smoother, more consistent animations while avoiding the geometric distortions and temporal artifacts found in previous methods like AniClipart. The experimental results showcase the advantages of FlexiClip in producing better animations across diverse clipart types.

This paper doesn't include Impact Statements, which are required by ICML. Therefore, I suggest rejection.

**Claims And Evidence:**

Most of the claims are proved via experiments and results. While there are some geometric distortions that occur in the results.

**Essential References Not Discussed:**

I'm not familiar with this area, so it's hard for me to write this part.

**Experimental Designs Or Analyses:**

Yes, the experiments looks sound to me, and this paper follows the experiment setup of an existing work.

**Methods And Evaluation Criteria:**

Yeah, the methods make sense to me.

**Other Comments Or Suggestions:**

None

**Other Strengths And Weaknesses:**

Strengths: The use of probability flow ODEs (pfODEs) provides a mathematically grounded way to ensure temporal smoothness.
The flow matching loss based on GFlowNet principles is a reasonable adaptation.

Weakness: Some results show noticeable distortions, particularly in human characters (e.g., hands, faces, and body parts deform unnaturally). While motion smoothness is analyzed, the paper lacks a quantitative evaluation of motion accuracy.

**Questions For Authors:**

None

**Relation To Broader Scientific Literature:**

I'm not familiar with this area, so it's hard for me to write this part.

**Theoretical Claims:**

None

---

> ### Author Rebuttal · Authors · 2025-03-25
>
> Thanks for your review we will include the following impact statement, basically we did'nt included the impact statement since we thought that it will count towards page limit during review process and can be added easily when accepted since 1 page extra will be given to submit the accepted paper to address the reviewer comments. Also, if you read this https://icml.cc/Conferences/2025/CallForPapers, that reviewer does not have authority to reject submission based on missing impact statement only program chairs can do that. Also there is this FAQ also there is this FAQ which clarifies that we did not desk-reject any papers for missing impact statements: https://icml.cc/Conferences/2025/PeerReviewFAQ.
>
> Please find the below the detailed impact statement for our work, we request you to please revise your acceptance score for this paper:
>
> The innovations in FlexiClip have significant implications for multiple domains:
>
> **Animation Industry**
> FlexiClip could streamline production processes for 2D animation, potentially reducing the labor-intensive nature of traditional animation while maintaining artistic quality. The technology could be particularly valuable for independent animators and small studios without access to extensive resources.
>
> **Digital Content Creation**
> As online platforms continue to prioritize dynamic content, FlexiClip's approach could enable creators to efficiently produce engaging animated content from static images, expanding creative possibilities for digital storytelling, marketing, and educational resources.
>
> **User Interface Design**
> The locality-preserving aspects of FlexiClip could inform new approaches to animated user interfaces, where maintaining visual coherence during transitions between states is crucial for user experience.
>
> **Broader Implications**
> Beyond its immediate technical contributions, FlexiClip represents a significant step toward bridging traditional animation principles with modern computational techniques. By addressing the intertwined challenges of temporal consistency and geometric integrity, the research sets a new standard for character animation that respects both the artistic qualities of the original image and the physical plausibility of movement.
>
> The paper "FlexiClip: Locality-Preserving Free-Form Character Animation" introduces transformative advancements in the field of clipart animation, addressing critical challenges such as maintaining visual fidelity and ensuring temporal coherence in character motion. FlexiClip achieves smooth, natural-looking animations while preserving the structural integrity of clipart images. Additionally, its integration with pre-trained video diffusion models through methods like Video Score Distillation Sampling (SDS) bridges traditional animation principles with cutting-edge machine learning frameworks.
>
> These innovations have far-reaching implications for the animation industry, enabling streamlined workflows for animators and reducing the labor-intensive nature of traditional animation processes. FlexiClip's ability to generate high-quality animations from static clipart images expands creative possibilities for digital content creation, marketing, and educational applications. Furthermore, its locality-preserving approach holds promise for enhancing animated user interfaces, improving transitions and user experiences.
>
> Beyond practical applications, FlexiClip sets a new benchmark in character animation by addressing intertwined challenges of temporal smoothness and geometric fidelity. Its robust framework not only advances artistic expression but also paves the way for future developments in computer graphics, animation technology, and related fields such as robotics and biomechanics. By redefining standards for seamless motion generation, FlexiClip establishes itself as a foundational contribution to the evolution of animation techniques in the digital age.

---

> > ### Comment · Reviewer_Ry9s · 2025-04-01
> >
> > I understand that ICML does not desk reject papers solely for missing impact statements, so I will adjust my score accordingly. That said, I noticed the authors did not respond to the weakness I raised earlier, and I’m wondering if there’s a particular reason for that.

---

> > > ### Author Response · Authors · 2025-04-01
> > >
> > > Thanks for your acknowledgement on Rebuttal and many thanks on reminding us about the weakness part you have asked in your review. Please find below the answer for your question on Motion Accuracy.
> > >
> > > The motion accuracy is precisely the indicator how well the animation follows the text prompt with valid motion trajectory and that's what we have evaluated with Subjective User Study presented in Table 3 where we have evaluated Text Video Alignment by getting the rating from the annotators and they are provided with the instructions to judge that the character should follow the valid and motion trajectory as per the prompt. There is no general open source model to do that and the valid trajectory requires the visual inspection and hence we believe the animations demonstration side by side with the text prompt should be evaluated manually by the annotators to get the fair judgement on this.
> > >
> > > We invite you to see our project page https://creative-gen.github.io/flexiclip.github.io/ with detailed animations side by side with the prompt to get a feel for the smooth and accurate animations.
> > >
> > > Hope this answer resolves your ask on Motion accuracy.

---

### Official Review · Reviewer_GNpb · 2025-03-14

**Overall Recommendation:** 4

**Summary:**

This paper addresses several key challenges in the the problem of clipart animation. To address the noise accumulation along the animation, the paper proses the novel concept of temporal Jacobians to correct the temporal noise. To ensure the smooth temporal transitions between frames, the paper proposes pfODE to model the temporal Jacobians. A novel flow matching loss is introduced to optimize the temporal noise. The result: high quality clipart animation with great temporal coherence. The experiments are quite comprehensive. The major technical claims from the paper are all well validated.

Update after rebuttal: after reading other reviewers' comments and the authors' rebuttal, my rating of acceptance remains unchanged.

**Claims And Evidence:**

I find that all the major technical claims are supported by intuitions and experimental validation.
Claim 1: the temporal Jacobians idea is novel and it is the base to a solution to temporal noise accumulation. To my knowledge, this novelty claim is true, and the effectiveness is validated in experiment Sec. 4.4  - removing this part leads to noticeable temporal artifacts.

Claim 2: the pfODE formulation and the flow matching loss enables temporal smoothness. This claim is supported by the change in several metrics after removing this loss from training (Sec. 4.4).

Claim 3: the proposed method suits multiple scenarios including multi-object, layered objects and rotation. These are demonstrated in experiment Sec. 4.5

Overall, I find this paper solid in making and validating the claims.

**Essential References Not Discussed:**

N.A.

**Experimental Designs Or Analyses:**

I appreciate the well-organized experiment section. The metrics are clearly defined and they are well-designed to reflect different aspects of animation -- the extent of the motion (AE, MV), temporal coherency (TC, DS) and visual identity preservation. The new method is in-depth compared with the most recent baseline, AniClipart (Sec. 4.3). The findings in this section well align with the motivations and claims in the paper introduction. User study further corroborates the superior perceptual quality of the new method. The ablation study is the key evaluation -- confirming that the three major components in the proposed method are effective and necessary.
In all, the experiment section is strong.

**Methods And Evaluation Criteria:**

Overall the method makes a lot of sense to me. The paper first identifies the key problem with previous clipart animation methods -- using rigid parameterization (such as the ARAP prior in AniClipart) for motion, and do not explicitly account for the noise accumulation over time. It is a great idea to decompose the deformation Jacobian into the spatial one and per-time-step corrective Jacobians, with the latter being optimized using flow matching loss -- treating the spatial Jacobian as the backward process that introduces temporal noise and the forward process reduces such noise. The formulation is elegant and effective.

As for evaluation, the paper compares with a number of recent methods on the same (clipart animation) or neighboring (video generation) tasks. The metrics are chosen to well reflect the major improvements brought by this paper -- temporal coherency, deformation smoothness, and the capability to preserve identity. These design make the evaluation results convincing.

**Other Comments Or Suggestions:**

I do not have other major comments.

**Other Strengths And Weaknesses:**

I find this paper very solid: practical problem, clever idea, and extensive experiments that validates all the major technical novolties. The paper is self-contained and well written. I'd recommend acceptance.

**Questions For Authors:**

Is the method applicable for 3D mesh deformation?

**Relation To Broader Scientific Literature:**

The paper proposes a novel and effective method for clipart animation from single image input. It can be of great interest to the computer graphics community as a novel alternative for automatic character animation.

**Theoretical Claims:**

The major theoretical claims (implicitly made) are that (1) The spatial + temporally corrective Jacobians are suitable representation for this task; (2) the probability flow ODE is suitable to model the evolution of the temporal Jacobians, and (3) the flow matching loss can effectively minimize the temporal noise. These theories are outside of my major research domain, but I tried to understand them reading through Sec. 2 and 3 and the analysis and derivations make sense to me. Would love to discuss and learn from other reviewers to give a solid assessment on this part.

---

> ### Author Rebuttal · Authors · 2025-03-26
>
> Many thanks for reviewing our paper and for your detailed and thoughtful evaluation. I truly appreciate the time and effort you put into this, as well as your unbiased rating of our work.
>
> Regarding your question about the 3D extension, FlexiClip can easily be adapted for 3D animation. This requires adding just one extra coordinate and learning the Bézier trajectory over it, which is straightforward for our approach. In fact, we plan to release the code with configurable 2D and 3D settings.
>
> To specifically address your question—yes, our method can be extended to 3D mesh deformation, and we will include the 3D extension in our code release.
>
> Let me know if you need any further refinements!

---

### Decision · Program_Chairs · 2025-05-01

**Decision:**

Accept (poster)

**Comment:**

The paper proposes an innovative framework FlexiClip to address the challenges of maintaining visual fidelity and temporal coherence in animated clipart.  The technical contributions include: (1) temporal Jacobians for incremental motion correction, (2) continuous-time modeling through probability flow ODEs, and (3) a flow matching loss for reducing temporal noise. The proposed FlexiClip outperforms previous methods, achieving better scores in visual identity and animation quality. While the technical contributions and potential impact of the work are acknowledged, there are concerns about the extent of visual improvements and the limited scope of the evaluation.


During the rebuttal phase, the authors have made efforts to address the reviewers' concerns during the rebuttal phase, particularly by clarifying the differences in their results compared to existing methods and explaining their evaluation methodology. They have also provided additional resources, such as a project page with detailed animations, to help reviewers better assess the quality of their results. These efforts lead to a consensus voting for the paper acceptance.


Given the positive feedback from all the reviewers and potentially impactful approach to animated clipart, the AC is happy to recommend accepting this paper for publication in ICML 2025. Congratulations! Please be aware that the authors are strongly encouraged to address the aforementioned points in their camera-ready version.